# Early Hemodynamic Changes following Surgical Ablation of the Right Greater Splanchnic Nerve for the Treatment of Heart Failure with Preserved Ejection Fraction

**DOI:** 10.3390/jcm11041063

**Published:** 2022-02-18

**Authors:** Piotr Gajewski, Marat Fudim, Veraprapas Kittipibul, Zoar J. Engelman, Jan Biegus, Robert Zymliński, Piotr Ponikowski

**Affiliations:** 1Institute of Heart Diseases, University Hospital, 50-556 Wroclaw, Poland; janbiegus@gmail.com (J.B.); robertzymlinski@gmail.com (R.Z.); ppponikowski@gmail.com (P.P.); 2Institute of Heart Diseases, Medical University, 50-556 Wroclaw, Poland; 3Duke Clinical Research Institute, Durham, NC 27710, USA; marat.fudim@googlemail.com; 4Division of Cardiology, Duke University Medical Center, Durham, NC 27710, USA; veraprapas.kittipibul@duke.edu; 5Coridea LLC, New York, NY 10018, USA; zengelman@gmail.com

**Keywords:** heart failure, HFpEF, greater splanchnic nerve ablation

## Abstract

Background: Permanent ablation of the right greater splanchnic nerve (GSN) has previously been demonstrated to improve quality of life and functional outcomes, as well as reduce abnormally high intracardiac filling pressures, in patients with heart failure with preserved ejection fraction (HFpEF) at 1, 3 and 12 months following the procedure. We hypothesize that hemodynamic changes that ensue from surgical right GSN ablation would be apparent as early as 24 h after the medical intervention. Methods and Results: This is a prespecified analysis of a single-arm, two-center, open-label study evaluating the effects of right GSN ablation via thoracoscopic surgery in HFpEF patients with pulmonary capillary wedge pressure (PCWP) ≥15 mmHg at rest or ≥25 mmHg with supine cycle ergometry. A total of seven patients (median age 67 years, 29% female) underwent GSN removal followed by invasive right heart catheterization within 24 h. GSN ablation resulted in a significant reduction in PCWP 24 h after the procedure compared to baseline for both 20 W exercise (baseline (28.0 ± 4.3 mmHg) to 24 h (19.6 ± 6.9 mmHg); *p* = 0.0124) and peak exercise (baseline (25.6 ± 2.4 mmHg) to 24 h (17.4 ± 5.9 mmHg); *p* = 0.0025). There were no significant changes in resting or leg-up hemodynamics. Conclusions: Permanent right GSN ablation leads to a reduction in intracardiac filling pressures during exercise, apparent as early as 24 h following the procedure.

## 1. Background

Heart failure with preserved ejection fraction (HFpEF) comprises about 50% of today’s heart failure population, and its incidence is constantly increasing [1,2]. Unlike heart failure with reduced ejection fraction (HFrEF), in HfpEF, there are no well-established drug therapies. Current clinical approaches focus on modifying risk factors and comorbidities to control symptoms in HFpEF [3,4]. The results of the EMPEROR-Preserved study published in 2021 indicate a new option for pharmacological treatment to reduce the combined risk of death from cardiovascular causes and hospitalization due to heart failure [5]. Preliminary evidence suggests that lowering exercise induces intracardiac pressures with the interatrial shunt procedure, yet the pivotal study results are pending.

The hallmark of HFpEF is exercise intolerance, which is manifested by exertional dyspnea or fatigue. Growing evidence shows that an uncontrolled hemodynamic response to exercise, as manifested by a rapid increase in intracardiac filling pressures (which usually return to baseline in the rest) can be responsible for this condition [6]. Volume redistribution, in addition to total body fluid retention, is increasingly being recognized as an important contributor of elevated intracardiac pressures and clinical congestion in heart failure [7]. The splanchnic venous reservoir plays a critical role in controlling the distribution of blood between stressed and unstressed compartments in the body [8]. In heart failure, there is a decreased capacity of the splanchnic vascular reservoir to buffer volume shifts in the body, leading to an abnormal rise in central pressures during exertion, even in the setting of normal hemodynamics at rest as commonly seen in patients with HFpEF [9]. Various interventions aimed at selectively affecting the splanchnic system to improve outcomes in patients with HF have been investigated, with specific focus on targeted modulation of the greater splanchnic nerve (GSN) [10]. The potential benefits of splanchnic nerve modulation in HF are believed to be related to sympathetically mediated improvement in vascular compliance and a decrease in inappropriately high intracardiac filling pressures at rest and especially with exertion [11].

Recently, the feasibility and safety of permanent right GSN ablation in HFpEF were examined in a small proof-of-concept study [12]. This study demonstrated that right GSN ablation in HFpEF was safe, with no adverse events related to the absence of the GSN for at least 12 months. Mechanistically, there was a significant reduction in intracardiac filling pressures during exercise right-heart catheterization at 1, 3, and 12 months after the procedure compared to baseline. Clinically, patients demonstrated significant improvement in quality of life and functional capacity following GSN ablation through 12-month follow-ups as compared to baseline. The early hemodynamic changes following GSN ablation have not yet been described. In this study, we sought to examine the changes in invasive hemodynamic measurements within 24 h following surgical GSN ablation in patients with HFpEF.

## 2. Methods

The study design and the primary results have been previously published [11]. Briefly, patients were enrolled in a single-arm, two-center, open-label, prospective study aimed at the feasibility of elective blockade of sympathetic signaling to the splanchnic circulation by surgical ablation of the right GSN (clinicaltrial.gov, NCT03715543). To be considered for enrollment, patients had to be ≥18 years of age with guideline-defined HFpEF, New York Heart Association (NYHA) functional class III/IV, and pulmonary capillary wedge pressure (PCWP) ≥15 mmHg at rest or ≥25 mmHg during exercise. The original study enrolled a total of 10 patients (from 15 patients screened) between June 2016 and July 2017. All patients underwent surgical ablation of the right GSN using a multi-port video-assisted thoracoscopic approach. Seven of the ten patients who recovered from the surgical intervention underwent repeat hemodynamic testing approximately 24 h after the original procedure.

The early clinical effectiveness of GSN ablation was assessed by examining changes in hemodynamic measurements obtained from invasive right heart catheterization approximately 24 h after the procedure compared with baseline. Central hemodynamic profiles (i.e., central venous pressure (CVP) and systolic pulmonary artery pressure (PAP-S), PCWP) were measured at rest, during leg-up maneuver, and during supine bicycle exercise. Supine bicycle exercise protocol was implemented by commencing at 20 watts (W) with 10 W increments every 90 s until the patient achieved maximum effort as defined by symptom-limiting dyspnea or fatigue. The same central hemodynamic measurements were taken after a five-minute recovery from the end of maximal exertion. Summaries within a visit are presented as mean ± standard deviation or median (Q1, Q3), unless otherwise noted, and change from baseline is presented as median (95% confidence interval [CI]). Hemodynamic data were compared using Wilcoxon Signed Rank test (SAS v9.4 for Windows, SAS Institute Inc., Cary, NC, USA). A *p* value < 0.05 was considered statistically significant.

## 3. Results

Baseline characteristics of seven enrolled patients are summarized in Table 1. Patients had a median age of 67 years, were 29% female and had high burden of comorbidities (86% with atrial fibrillation and 71% with arterial hypertension). All patients were on diuretics and had a high utilization of anti-hypertensive/HF medications. At 24 h after undergoing surgical right GSN ablation, there was no significant change in resting CVP (baseline (9.9 ± 5.0 mmHg) to 24 h (7.43 ± 2.99 mmHg); *p* = 0.199), resting PAP-S (baseline (37.0 ± 8.7 mmHg) to 24 h (37.1 ± 8.7 mmHg); *p* = 0.898) or resting PCWP (baseline (15.7 ± 2.7 mmHg) to 24 h (14.9 ± 3.5 mmHg); *p* > 0.999). In contrast, there was a significant reduction in PWCP with 20 W (baseline (28.0 ± 4.3 mmHg) to 24 h (20.7 ± 6.0 mmHg); *p* = 0.0124) and peak exercise (baseline (25.6 ± 2.4 mmHg) to 24 h (18.6 ± 5.4 mmHg); *p* = 0.0025) (Figure 1). There was a non-significant trend toward reduction in PCWP with leg-up (baseline (21.9 ± 3.6 mmHg) to 24 h (17.6 ± 4.2 mmHg); *p* = 0.0714).

The early (24 h) hemodynamic changes after the GSN ablation correlated well with long-term post-procedure hemodynamic adaptations. There was a similar statistically significant reduction in PCWP with leg-up (16.9 ± 3.8 mmHg; *p* = 0.0278) and 20 W exercise (20.3 ± 6.4 mmHg; *p* = 0.0217) one year after GSN resection compared to baseline. Although the 24 h PCWPs were often lower than those measured after one year, there was no statistically significant differences between the groups for either rest (*p* = 0.379), leg-up (*p* = 0.745), or 20 W exercise (*p* = 0.843).

Changes were also observed in the intracardiac pressure during the recovery phase of exercise. In all patients, there was a decrease in recovery PCWP in the first 24 h after the procedure vs. baseline (15.6 ± 4.7 vs. 20.4 ± 5.0 mmHg, *p* < 0.027). A similar trajectory was observed in the annual follow-up with mean recovery PCWP of 17.3 ± 9.1 mmHg, although this did not meet statistical significance compared to baseline (*p* = 0.31).

## 4. Discussion

The persistent hemodynamic and clinical benefits of permanent GSN ablation have been described previously [12]; herein, we describe for the first time in HFpEF patients undergoing permanent right GSN ablation that hemodynamic improvements occur as early as 24 h after the procedure. These results of permanent GSN ablation in HFpEF support the mechanistic insights and immediate hemodynamic benefits seen with temporary GSN modulation in both decompensated hospitalized HF (splanchnic HF-1) [13] and chronic ambulatory HF (splanchnic HF-2) [14]. As opposed to these studies, which enrolled predominantly HFrEF patients (91% HFrEF in splanchnic HF-I and 93% HFrEF in splanchnic HF-II), the current study exclusively enrolled patients with HFpEF. The consistent and favorable effects of GSN modulation on hemodynamics in the HFpEF phenotype is encouraging, as this group historically does not derive the same therapeutic benefits from HFrEF treatments.

Similar to follow-ups at 1, 3, and 12 months [12], the early hemodynamic changes following permanent GSN ablation appear to be more prominent during exercise than at rest, and they failed to reach statistical significance at 24 h follow-up. Conversely, a significant reduction in resting filling pressures was observed in splanchnic HF-I and splanchnic HF-II trials. The greater administration of supporting intravenous fluid and blood product during the surgical procedure as opposed to temporary block procedures may explain, in part, some of the observation differences in resting pressures. Despite this, patients still exhibited significant improvement in exercise hemodynamics, signifying the promising benefits of GSN ablation even in the setting of increased fluid retention. The observed difference may potentially be explained by the incremental effect of bilateral block over unilateral ablation. Nevertheless, the differential effects of GSN ablation on reducing filling pressures only during exercise highlight the important role of the splanchnic nervous system in reducing stressed blood volume that underlies exercise intolerance in HFpEF.

The consistency of hemodynamic and clinical benefits seen across these studies speaks to the importance of the splanchnic vascular reservoir in the pathophysiology of heart failure independent of ejection fraction. These encouraging results, together with a reasonable safety profile of GSN modulation [10], pave the way for larger randomized controlled studies needed to show long-term benefits, tolerability, and safety in HF, as well as the best technical approach for GSN modulation.

## 5. Limitations

This study has some limitations, in addition to what was described in the original study, that need to be considered. First, not all patients underwent right heart catheterization at 24 h following surgical GSN ablation at the discretion of the treating physician. This subjects the results to possible selection bias in that only patients who recovered well enough for the catheterization and exercise could have derived greater benefits from the procedure. Second, clinical variables (e.g., weight, NT-proBNP) other than invasive hemodynamic measurements were not recorded at 24 h and were not available for comparison.

## 6. Conclusions

From the retrospective analysis of the single-arm, open-label, prospective study, a reduction in intracardiac filling pressures during exercise was observed as early as 24 h following permanent right GSN ablation in patients with HFpEF.

## Figures and Tables

**Figure 1 jcm-11-01063-f001:**
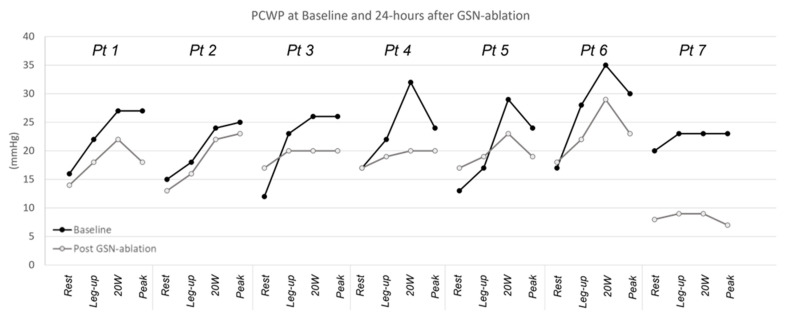
Resting and exercise pulmonary capillary wedge pressure. Abbreviations: GSN, greater splanchnic nerve; PCWP, pulmonary capillary wedge pressure.

**Table 1 jcm-11-01063-t001:** Baseline demographic characteristics (n = 7).

Age ± SD (years)	67 ± 11
Female (%)	2 (29)
Body Mass Index, median (Interquartile range) (kg/m^2^)	30 (29–35)
Comorbidities	
History of Atrial Fibrillation (%)	6 (86)
Hypertension (%)	5 (71)
Diabetes (%)	3 (43)
Coronary Artery Disease (%)	4 (57)
Previous Myocardial Infarction (%)	3 (43)
Left Ventricular Ejection Fraction ± SD (%)	54 ± 7
NYHA Class I/II/III/IV (%)	0/0/100/0
Arterial Blood Pressure, systolic/diastolic ± SD (mmHg)	126/80 ± 15/14
Resting Heart Rate (beats/min)	80 ± 9
NT-proBNP, median (Interquartile range) (pg/mL)	1220 (51–2797)
Creatinine, median (Interquartile range) (mg/dL)	1.1 (1.0-1.5)
eGFR ± SD (mL/min/1.73 m^2^)	63 ± 16
Heart failure or anti-hypertension medication	
Loop Diuretic (%)	7 (100)
ACEi or ARB (%)	6 (86)
Beta-Blocker (%)	6 (86)
MRA (%)	6 (86)
CCB (%)	2 (29)
Other vasodilators (%)	1 (14)

Abbreviations: NYHA, New York Heart Association; ACEi, angiotensin-converting enzyme inhibitors; ARB, angiotensin receptor blockers; MRA, mineralocorticoid receptor antagonists; CCB, calcium channel blockers; NT-proBNP, N terminal pro-natriuretic peptide; eGFR, estimated glomerular filtration rate. Results are presented as mean ± standard deviation (SD) unless otherwise specified.

## Data Availability

The data presented in this study are available on request from the corresponding author. The data are not publicly due to privacy restrictions.

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
