# Peer review of "Early Hemodynamic Changes following Surgical Ablation of the Right Greater Splanchnic Nerve for the Treatment of Heart Failure with Preserved Ejection Fraction"

_jcm, 2022, doi:10.3390/jcm11041063_

Round 1

Reviewer 1 Report

Gajewski et al. provide insight to a novel field of heart failure therapy.

The quality of writing and use of English language is fine. Methods are clearly described, results and discussion focus on the results presented without too much speculation.

The piece of work definitely has merit. Although the number of patients is low, the submission as a communication is adequate and therefore well chosen.

To my mind, there are only minor corrections necessary. The introduction would benefit from a short discussion (2,3 sentences) about medicamentous treatment options of HFpEF and maybe also about other interventional (IASD) options. Furthermore, the text box of Figure 1 overlays the figure itself partly. This should be fixed.

Author Response

Thank you very much for your comment and valuable advice. I fully agree with You.

We have added information on current treatments for HFpEF to our manuscript.

„Unlike heart failure with reduced ejection fraction (HFrEF), in HFpEF we do not have well established drug therapies. Current clinical approach focuses on modifying risk factors and comorbidities to control symptoms in HFpEF [3,4].  The results of the EMPEROR-Preserved study published in 2021 indicate a new option for pharmacological treatment to reduce the combined risk of death from cardiovascular causes and hospitalization due to heart failure [5].  Preliminary evidence suggests lower exercise induced intra cardiac pressures with the interatrial shunt procedure, yet the pivotal study results are yet pending. “

Due to the small sample size, we also changed Figure 1 and presented the data of each patient separately

Reviewer 2 Report

it is a good work. i have some concerns

  1. the selected patients have high BMI, could  the author give some explainations?
  2. as a small sample study, it will be better to show every subjects in figure 1 comparerd with mean ± standard deviation.

Author Response

Thank you very much for your comment and valuable advice.

  1. The study protocol did not define body weight ranges that could exclude the patient from the study. Some patients were obese, which is one of the most common comorbidities in this groups of patients (prevalence of up to 50%).
  2. Due to the small sample size, we changed Figure 1 and presented the data of each patient separately

Reviewer 3 Report

In the present study the authors report near-immediate (within 24h) effects of permanent splanchnic nerve block (SNB) on intracardiac pressures at rest and with excercise in patients with HFpEF.
The manuscript under review is based on a previously published study that examined the medium-term effects (follow-up of 1 year) of the procedure. A lasting effect was seen with a reduction in PCWP with excercise at 1 year which was already significant at the 1-month follow-up.
At the same time it has been shown by the same group that hemodynamic effects with (temporary) SNB can be seen as early as 1 hour after the procedure and there is no reason to expect that this would be fundamentally different in permanent SNB.
The current paper is well written and reports previously unpublished data, but the findings are not exactly novel. The entire research group is to be commended for their important work and stringent research programme in the field of SNB, but it is unclear which substantial new findings of clinical relevance are added by the the manuscript under review.

Author Response

Thank you very much for your comment.
Unlike prior investigations, in this study we used a surgical unilateral approach. Prior acute studies looked at an anesthesiological block without direct visualization of the nerves. In most cases a bilateral block was performed using local anesthetics. Our ability to see acute changes with a unilateral approach alone is reassuring. It also builds on the evidence that pressures are an acute surrogate of technical success, thus we believe our manuscript is of interest to the cardiovascular community.